# Smell—Adding a New Dimension to Urinalysis

**DOI:** 10.3390/bios10050048

**Published:** 2020-05-05

**Authors:** Eva H. Visser, Daan J. C. Berkhout, Jiwanjot Singh, Annemieke Vermeulen, Niloufar Ashtiani, Nanne K. de Boer, Joanna A. E. van Wijk, Tim G. de Meij, Arend Bökenkamp

**Affiliations:** 1Department of Pediatric Gastro-Enterology, Emma Children’s Hospital, Amsterdam UMC, Vrije Universiteit Amsterdam, 1081 HV Amsterdam, The Netherlands; d.berkhout@amsterdamumc.nl (D.J.C.B.); j.singh@amsterdamumc.nl (J.S.); t.demeij@amsterdamumc.nl (T.G.d.M.); 2Department of Pediatrics, Emma Children’s Hospital, Amsterdam UMC, Vrije Universiteit Amsterdam, 1081 HV Amsterdam, The Netherlands; a.vermeulen@amsterdamumc.nl; 3Department of Pediatrics, OLVG Oost, 1091 AC Amsterdam, The Netherlands; niloufar13@gmail.com; 4Department of Gastroenterology and Hepatology, Amsterdam Gastroenterology and Metabolism Research Institute, Amsterdam UMC, Vrije Universiteit Amsterdam, 1081 HV Amsterdam, The Netherlands; khn.deboer@amsterdamumc.nl; 5Department of Pediatric Nephrology, Emma Children’s Hospital, Amsterdam UMC, Vrije Universiteit Amsterdam, 1081 HV Amsterdam, The Netherlands; jae.vanwijk@amsterdamumc.nl (J.A.E.v.W.); a.bokenkamp@amsterdamumc.nl (A.B.)

**Keywords:** urinary tract infection, electronic nose, bacterial growth culture, volatile organic compounds

## Abstract

Background: Urinary tract infections (UTI) are among the most common infections in children. The primary tool to detect UTI is dipstick urinalysis; however, this has limited sensitivity and specificity. Therefore, urine culture has to be performed to confirm a UTI. Urinary volatile organic compounds (VOC) may serve as potential biomarker for diagnosing UTI. Previous studies on urinary VOCs focused on detection of UTI in a general population; therefore, this proof-of-principle study was set up in a clinical high-risk pediatric population. Methods: This study was performed at a tertiary nephro-urological clinic. Patients included were 0–18 years, clinically suspected of a UTI, and had abnormal urinalysis. Urine samples were divided into four groups, i.e., urine without bacterial growth, contamination, colonization, and UTI. VOC analysis was performed using an electronic nose (eNose) (Cyranose 320^®^) and VOC profiles of subgroups were compared. Results: Urinary VOC analysis discriminated between UTI and non-UTI samples (AUC 0.70; *p* = 0.048; sensitivity 0.67, specificity 0.70). The diagnostic accuracy of VOCs improved when comparing urine without bacterial growth versus with UTI (AUC 0.80; *p* = 0.009, sensitivity 0.79, specificity 0.75). Conclusions: In an intention-to-diagnose high-risk pediatric population, UTI could be discriminated from non-UTI by VOC profiling, using an eNose. Since eNose can be used as bed-side test, these results suggest that urinary VOC analysis may serve as an adjuvant in the diagnostic work-up of UTI in children.

## 1. Introduction

Urinary tract infections (UTI) are among the most common bacterial infections in children worldwide [1]. The prevalence of UTI in children is influenced by a wide variety of factors, including age, gender, race, and circumcision status [1]. The highest prevalence of UTI in children is observed in boys younger than one and in girls around the age of four [1]. higher prevalence of UTI is observed in children with renal or urological diseases, particularly in case of congenital anomalies of kidney and urinary tract (CAKUT) [2]. In this specific population, early diagnosis and treatment of UTI is of utmost importance to prevent renal scaring [3].

The primary diagnostic method to detect UTI is by urinalysis. Urinalysis quantitatively measures urinary leukocytes (white blood cells as a marker of an inflammatory response to infection), nitrite, erythrocytes, and an estimation of bacterial content by microscopy [4]. The latter increases the diagnostic accuracy of urine testing [5]. In daily practice, this method is often replaced by the dipstick test, a bedside test measuring leucocytes, nitrite, and erythrocytes by dry chemistry. This screening has, however, limited sensitivity and specificity [6]. Therefore, urine culture is required to make a definite diagnosis of UTI. However, it takes 24 h to confirm the presence of bacteria by urine culture and at least 1 to 2 days to identify the type of bacteria and determine antibiotic susceptibility [4]. This may result in delayed initiation of therapy or on the contrary, in unnecessary antibiotic treatment.

In the lay literature, a foul smell of urine is considered as a symptom of UTI; however, this subjective symptom relying on the human olfactory system has not been confirmed scientifically [7,8]. A potential novel diagnostic biomarker for detection of UTI are urinary volatile organic compounds (VOCs). VOCs are chemicals in volatile state at ambient temperature, which are produced during physiological, pathophysiological, and microbial processes and which cause the smell of a gaseous mixture. VOC patterns can be detected in the headspace of different bodily excretions, such as sweat, feces, or urine [9]. Various techniques have been developed for the detection of VOCs, which can roughly be divided into two main groups: electronic devices, using pattern-based recognition algorithms, and chemical analytical techniques, which allows for detection of molecules on individual level [10]. The former is also known as an electronic nose (eNose) and has been used in previous studies to determine VOC patterns. Since VOCs are produced by host and microbiota and during host-microbiota response, they have increasingly been evaluated as a potential diagnostic biomarker for diseases in which the pathophysiology is linked to microbial alterations, like pulmonary aspergillosis, *Clostridium difficile* infections, and inflammatory bowel disease [11,12,13].

Since UTI are caused by microbes, urinary VOCs might hypothetically serve as a biomarker for UTI. Previous studies using an eNose have shown promising results [14,15]. Notably, an eNose even seemed to allow for discrimination between different pathogens causing UTI [15,16]. By bed-side detection of the causative pathogen, a more specific antibiotic therapy could consequently be started at an early stage. The previous studies on urinary VOCs focused on detection of UTI in a general population, using mostly anonymous urine samples, while the potential of VOCs as biomarker of UTI in a high-risk pediatric population has not been explored yet.

Therefore, this clinical proof-of-principle study was set up to assess whether urinary VOC analysis is a potential tool for the diagnosis of UTI in children, at a tertiary nephro-urological clinic.

## 2. Materials and Methods

### 2.1. Participants and Procedures 

This prospective, proof-of-principle study was performed from November 2016 until September 2017 at the Department of Pediatric Nephrology at Amsterdam University Medical Center, Netherlands. All patients between 0 and 18 years of age who were suspected of having a UTI based on clinical complaints and an abnormal urinalysis were eligible to participate. Patients with normal urine dipstick were excluded. Information on history and use of medication was obtained from each patient’s medical file. 

The study was approved by the Institutional Review Board of Amsterdam University Medical Center, location Vumc (reference number 2016.104).

### 2.2. Measures

#### 2.2.1. Urine Samples

Urine was collected by midstream collection (to eliminate potential contamination of urine by bacteria in the urethra) or urinary bladder catheterization (UBC, less prone for contamination than a midstream urine) at the outpatient clinic. The minimum sample volume was 6 mL. Urinalysis was performed using the dipstick method (Multistix^®^ 10 SG on a Clinitek Status+^®^ analyzer, Siemens Healthcare, The Hague, The Netherlands). Abnormal urinalysis was defined as a positive result for nitrite and leukocytes [17]. If an abnormal urinalysis was found the remaining urine was divided into three separate test tubes, one for urine culture and two for eNose analysis. All samples for eNose diagnostics were marked, directly frozen, and stored at −20 °C until further handling. 

#### 2.2.2. Electronic Nose Analysis

eNose analysis was performed using the Cyranose 320^®^ VOC analyzer (Smiths Detections, Pasadena, CA, USA). The Cyranose 320^®^ is a handheld chemical vapor analyzer, which contains a nanocomposite array comprising 32 polymer sensors, each having a different coating. When exposed to a gaseous mixture, the polymer sensors swell due to competitive interactions with the VOCs. This volume increase results in an increase in electrical resistance. Individual VOCs interact with multiple sensors and each sensor is influenced by multiple VOCs. This results in 32 resistance alterations which are combined in the VOC profile, also referred to as the smellprint. The VOC profile can then be used to differentiate clinical groups by pattern recognition analysis [18]. 

All urinary VOC analyses were performed in a single batch. Maximum storage time of the collected samples was one year. Samples were thawed to room temperature to increase the concentration of headspace VOCs. The thawing process took approximately 30 min. All samples were analyzed in random order. Two needles were pierced into the cap of the sealed vacutainer and connected to the airtight system of the eNose. The actual measurement was performed by letting the urine VOCs in the headspace pass an array of 32 sensors for 60 s. Sensors were purged for 90 s after each measurement to remove any remaining urine VOCs. 

Urine culture was performed at the microbiology laboratory of Amsterdam University Medical Center using standard procedures. UTI was defined as a urine culture with ≥10^5^ colony forming units (CFU) per milliliter causative of UTI. Such a high bacterial load is commonly used to distinguish bacteriuria originating in the bladder and the upper urinary tract from contamination originating from the urethra or the skin where the number of bacteria is much lower. Samples not meeting the criterion of UTI were divided into three groups: (i) contaminated urine culture, i.e., one or more bacteria species with 10^3^–10^4^ CFU/mL in a clean-catch urine, (ii) colonization, i.e., one or more bacteria species with 10^3^–10^4^ CFU/mL in a UBC sample, and (iii) urine without bacterial growth, i.e., less than 10^3^ CFU/mL. For the analysis, we compared UTI vs. urine without bacterial growth, UTI vs. no UTI (i.e., i + ii + iii) and urine with bacterial growth (i.e., UTI + i + ii) vs. urine without bacterial growth.

### 2.3. Statistical Analyses

The eNose contains 32 sensors which were recombined in a set of four principal components by principal component analysis, which were compared using an independent *t*-test. Discriminating principal components (PCs) were then used in a supervised canonical discriminant analysis (CDA), after which they were internally cross validated by means of leave one out method. Per variable of interest sensitivity and specificity were determined. For the discrimination between samples, scatterplots were created for each variable of interest. Axes depict two orthogonal linear recombinations of the raw sensor data by means of PC analysis; individual VOC profiles are illustrated as marked points. The intersection of the lines deriving from the individual profiles demonstrates the mean VOC profile of this specific variable of interest.

Statistical Package for the Social Science software version 21 (SPSS Inc., Chicago, IL, USA) was used for all statistical analysis. Demographical data are presented as frequencies or medians [range]. Performance of the eNose was assessed in terms of sensitivity and specificity and by calculating the area under the receiver-operating characteristic (ROC) curve. A *p*-value of <0.05 was considered statistically significant.

## 3. Results

### 3.1. Patient Characteristics

A total of 39 urine samples were included from 38 patients. Twelve samples met the definition of a UTI, while 27 did not (14 urine without bacterial growth, 4 colonization, 9 contamination). Table 1 summarizes the clinical characteristics of the different subgroups. In UTI samples, *E. coli* was the most frequent pathogen, present in 75%. All pathogens in urine with bacterial growth are shown in Table 2.

Comorbidities of the patients, in particular anatomical anomalies of the kidneys and urinary tract are detailed in Table 3.

### 3.2. VOC Analysis

Table 4 shows results of urinary VOC analysis comparing the different subgroups. VOC analysis allowed for discrimination between UTI and no-UTI samples. The area under the ROC curve was 0.70 (*p* = 0.048) with a sensitivity of 0.67 and a specificity of 0.70. Urine without bacterial growth could be discriminated from urine with bacterial growth (AUC 0.71; *p* = 0.033, sensitivity 0.64, specificity 0.79). The diagnostic accuracy of VOCs improved when comparing UTI versus urine without bacterial growth (AUC 0.80; *p* = 0.009, sensitivity 0.79, specificity 0.75).

## 4. Discussion

In this proof-of-principle study, we tested the performance of urinary VOC profiles for the diagnosis of UTI in children with a history of renal or urological diseases suspected of UTI and with an abnormal urinary dipstick. In this population, *E coli* was the most common causative pathogen of UTI, which is in line with the general pediatric population [19]. A wide variety of different pathogens were found (Table 2), reflecting the tertiary care nephro-urological population with a high prevalence of anatomical anomalies and use of prophylactic antibiotics. We observed that an eNose could discriminate UTI from non-UTI samples with modest accuracy, but statistically significant. Analysis of VOC showed best performance in the discrimination between UTI and urine without bacterial growth. Figure 1 visualizes these outcomes, with the individual VOC profiles of the different variables in two clouds which touch, indicating that sensitivity and specificity are not 100%. Still, ROC analysis indicates that the average VOC profiles are significantly different.

A few studies have explored the potential of urinary VOC profiling as a diagnostic biomarker for UTI in adult patients, albeit without detailed information about medical history or use of prophylactic antibiotics. Pavlou et al. compared VOC profiles to urine cultures using a gas-sensor array to analyze anonymous mid-stream urine samples from patients with a clinical diagnosis of acute uncomplicated UTI. In two case-series, they respectively examined 25 and 45 urinary samples, in the second half of the last series they correctly classified 18 of the 19 samples as UTI [14]. Roine et al. compared urinary VOC profiles between culture positive and culture negative urine samples by an eNose device in a total of 101 anonymous urine samples. They reported a sensitivity of 95% and specificity of 97% for the discrimination between sterile and non-sterile samples. Additionally, they were able to identify the causative organism (i.e., *E. coli*; *Staphylococcus saprophyticus*; *Klebsiella* spp.; *Enterococcus faecalis*) with a sensitivity of 95% and specificity of 96% [15]. Kodogiannis and Wadge tested 45 urine samples from randomly selected admitted patients suspected of a UTI, also using an eNose. They analyzed the VOC profiles by artificial intelligence. When comparing the VOC-profiles with urine culture, they found that artificial intelligence was able to differentiate bacteria causing a UTI [20]. 

The discriminative accuracy of urinary VOCs to detect UTI observed in our study was lower compared to previous studies. A possible explanation is that in most studies performed so far, UTI samples were compared to those of healthy controls. In contrast, in this study an intention-to-diagnose group was included, consisting of children clinically suspected of UTI and with abnormal urinalysis. Furthermore, all included children in this study had an extended medical history, and, consequently many of them used medication for a wide variety of indications. This heterogeneity in patient characteristics and medication causes an increased divergence in measured VOC patterns, which may influence diagnostic accuracy to detect UTI. Future studies should focus on identification of UTI-specific VOCs by means of gas chromatography–mass spectrometry (GC-MS), this may allow for development of disease-specific eNose sensors, increasing diagnostic accuracy.

Reported data underscore the potential of urinary VOC profiling as a diagnostic biomarker for UTI. Still, urinalysis is hampered by the presence of bacteria in low concentration in the urethra and under the prepuce or skin leading to contamination of urine samples. This is the reason why bacterial load is quantified when culturing urine, and why low concentrations (i.e., less than 10^5^ CFU/mL) are considered as contamination. This applies even more to children with continence issues and a narrow prepuce in uncircumcised boys before adolescence, making a mid-stream urine collection less reliable than in adults and prompted us to initiate this study in a clinically relevant setting in children.

In children suspected of UTI, urinalysis by dipstick is the first diagnostic test screening for the presence of leucocytes or nitrite [5]. While this test can be used to exclude UTI, it has limited specificity [6]. Therefore, confirmation by urine culture is obligatory in children [5], but takes at least one to two days before a (negative) result becomes available. In the clinical setting this raises the question whether antibiotics should be started immediately or treatment should be withheld until confirmation by urine culture, particularly in a population with an extended nephrological population, in which recurrent UTI may further impair the renal function. Here, the eNose, like urine microscopy for the presence of bacteria [5], may have an (adjuvant) diagnostic role in identifying samples without bacteriuria thus excluding a UTI, particularly since we observed that the performance of VOC analysis was optimal in discriminating UTI samples from urine samples without bacterial growth. The finding that the diagnostic accuracy of VOC analysis increased when comparing UTI to urine without bacterial growth, as compared to UTI vs non-UTI (including samples with contamination and colonization), suggests that urinary VOCs are, as expected, mainly influenced by the presence of bacteriuria. 

In patients with a negative eNose outcome, antibiotics may be withheld, and urine culture may be omitted while in the remainder, urine culture still needs to be done to confirm the diagnosis and also to determine antibiotic resistance. 

The strength of the present study is the set-up in a clinically relevant setting and a strict definition of UTI which also takes urine contamination and colonization into consideration. This study has several limitations: (i) The sample size of this proof-of-principle trial was small, limiting statistical power and the possibility to study interactions, such as the effect of antibiotic prophylaxis or other medication and anatomic abnormalities on the performance of the eNose. Also, the number of UTI samples was too low to assess whether the Cyranose 320^®^ can accurately distinguish bacteria-specific VOC patterns as suggested in other studies. (ii). All samples were analyzed in one batch, in randomized order and on the same day to eliminate potential environmental confounding factors on VOC outcome such as humidity, temperature, or other odors. This study setting does not represent the clinical setting and in future experiments urine should be analyzed twice, i.e., immediately and in batch to assess potential bias by storage and sampling conditions. (iii) Our patient population was limited to older children being able to produce a mid-stream urine and to children undergoing a UBC. In infants and younger children urine screening is often performed in samples collected by urine bags. This is associated with a higher proportion of contaminated urines [17]. Therefore, future studies should also include samples collected using a urine bag. (iv) We excluded samples with a negative urine dipstick. Therefore, asymptomatic bacteriuria [21], which is considered a benign condition has not been assessed. This also applies to malignancy patients with agranulocytosis who may have UTI in the absence of pyuria [22]. 

In conclusion, our data suggest that VOC profiling of urine samples using an eNose device may be a useful (complementary) technique in the diagnostic work-up of UTI in children, in particular in identifying patients with an abnormal dipstick who do not have a UTI. Further studies are needed in a larger cohort spanning the entire pediatric age spectrum with direct use of the eNose. Identification of individual urinary VOCs responsible for the differences in VOC profiles between UTI and non-UTI in children, using chemical analytical techniques, may allow for development of tailored eNose sensors, which may increase the diagnostic accuracy to detect UTI in clinical practice. 

## Figures and Tables

**Figure 1 biosensors-10-00048-f001:**
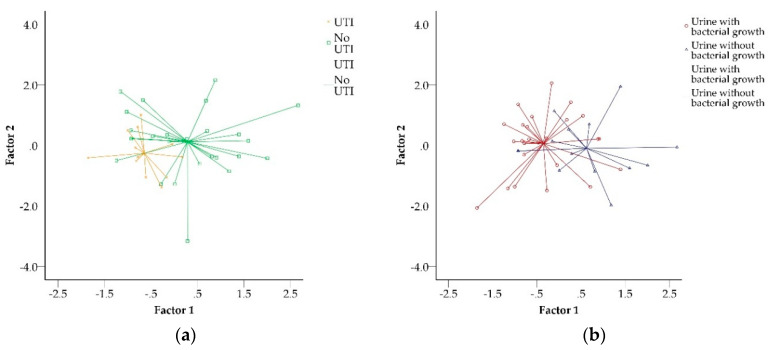
Scatterplots of principal component analysis; Scatterplot for the discrimination of a chemical fingerprint as analyzed by electronic nose (eNose), based on urine samples with urinary tract infection (UTI) or without UTI (**a**), and urine with and without bacterial growth (**b**). Axes depict two orthogonal linear recombinations of the original 32 sensor data, designed to capture the highest amount of original data variance by means of principal component analysis. These variables are called factors. Individual volatile organic compound (VOC) profiles are illustrated as marked dots. The intersection of the lines deriving from the individual profiles shows the mean VOC profile.

**Table 1 biosensors-10-00048-t001:** Patient characteristics.

	Urinary Tract Infection (*n* = 12)	No Urinary Tract Infection			Total (*n* = 39)
		Contamination (*n* = 9)	Colonization (*n* = 4)	Urine without Bacterial Growth (*n* = 14)	
		*Median (Range)*			
**Age (years)**	10.5 (0–17)	12.0 (2–18)	13.0 (11–15)	10.5 (0–18)	11.0 (0–18)
		*% (n)*			
**Male gender**	16.7% (2)	11.1% (1)	75.0% (3)	42.9% (6)	30.8% (12)
**Congenital anomalies of kidney and urinary tract**	41.7% (5)	44.4% (4)	75.0% (3)	35.7% (5)	43.5% (17)
**Use of prophylactic antibiotics**	33.3% (4)	33.3% (3)	50% (2)	28.5% (4)	33.3% (13)
Nitrofurantoin	16.7% (2)	11.1% (1)	25.0% (1)	7.1% (1)	12.8% (5)
Trimethoprim/sulfamethoxazole	8.3% (1)	11.1% (1)	25.0% (1)	21.4% (3)	15.4% (6)
Trimethoprim	8.3% (1)	11.1% (1)	0.0% (0)	0.0% (0)	5.1% (2)
**Bladder catheterization**	16.7% (2)	11.1% (1)	100.0% (4)	7.1% (1)	20.5% (8)

**Table 2 biosensors-10-00048-t002:** Spectrum of urinary pathogens.

	Urinary Tract Infection (*n* = 12)	Contamination (*n* = 9)	Colonization (*n* = 4)
***Escherichia coli***	75.0% (9)		
***Citrobacter* species**	16.7% (2)		
***Enterococcus faecalis* plus *Escherichia coli***	8.3% (1)		
**Mixed bacteria**		77.8% (7)	
***Enterococcus faecalis***		22.2% (2)	50.0% (2)
***Proteus mirabilis***			25.5% (1)
***Klebsiella pneumoniae* plus *Proteus mirabilis***			25.5% (1)

**Table 3 biosensors-10-00048-t003:** Comorbidities of all participants.

Urinary Tract Infection (*n* = 12)	No Urinary Tract Infection		
	*Contamination* (*n* = 9)	*Colonization* (*n* = 4)	*Urine without Bacterial Growth* (*n* = 14)
Hydronephrosis with megaureter Vesicoureteral reflux and kidney dysplasia Duplex system with ureterocele and vesicoureteral reflux Unilateral renal agenesis; Femur fibula ulna syndrome, constipation Chronic hypertensive kidney failure Purpura Henoch-Schönlein nephritis Nephrotic syndrome Spina bifida with neurogenic bladder Neurogenic bladder due to paraplegia caused by spinal fracture Diffuse pons glioma Embryonal rhabdomyosarcoma, ureter reimplantation; CKD stage 2 Recurrent UTI, dysfunctional voiding	Unilateral renal agenesis; dysfunctional voiding Duplex system with bladder diverticulum and vesicoureteral reflux Systemic lupus erythematodus Nephrotic syndrome Recurrent UTI, dysfunctional voiding Renovascular hypertension Persisting proteinuria Familial hematuria Prune-belly syndrome with vesicoureteral reflux, CKD stage 2–3	Posterior urethral valves, unilateral renal agenesis, iliovesicoplasty Posterior urethral valves with vesicoureteral reflux Spina bifida with neurogenic bladder Spina bifida with neurogenic bladder, iliovesicoplasty, bladder stone	Ureteropelvic junction obstruction Ureteropelvic junction obstruction Duplex system Unilateral multicystic dysplastic kidney Purpura Henoch-Schönlein nephritis Membranoproliferative glomerulonephritis, dysfunctional voiding VACTERL association, unilateral renal agenesis VACTERL association; neurogenic bladder Spina bifida with neurogenic bladder IgA nephropathy Familial hematuria Acute lymphatic leukemia Familial hematuria Recurrent UTI

**Table 4 biosensors-10-00048-t004:** Performance in distinguishing UTI from other conditions.

	Area under the Curve (AUC)	*p* Value	Sensitivity	Specificity
UTI (12) vs no UTI (27)	0.70	0.048	0.67	0.70
UTI (12) vs urine without bacterial growth (14)	0.80	0.009	0.79	0.75
Urine with bacterial growth (25) vs urine without bacterial growth (14)	0.71	0.033	0.64	0.79

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
