# Peer review of "Smell—Adding a New Dimension to Urinalysis"

_biosensors, 2020, doi:10.3390/bios10050048_

Round 1
Reviewer 1 Report
The manuscript describes a clinical study related to urinary tract infections (UTI) and the application of commercially available electronic nose, which is used without any modification.
The data presentation is uncomplete, at least the given figures can not be understood without legend.
The tables are presented in a sloppy style (please use a better readable format), and needs a more comprehensive discussion.
Reviewer 2 Report
Summary: The authors have presented a clinically relevant study of the urinary tract infection (UTI) using the primary detection method of urinalysis. Urinalysis although commonly practiced using the dipstick test to measure leucocytes, nitrite, and erythrocytes by dry chemistry, suffers from limited sensitivity and specificity. UTI is caused by microbes and produces several volatile organic compounds (VOCs) that can serve as a biomarker for early detection and diagnostic. In this study, an electronic nose (eNose) has been used to measure samples from young patients (0-18 years), clinically suspected of a UTI and had abnormal urinalysis results. VOC analysis using eNose discriminated between UTI and non-UTI samples. The accuracy was higher when VOC analysis was compared for urine without bacterial growth versus with UTI. The authors have included all relevant references and indicated their shortcomings. The authors need to include more experimental evidence to support their claims and demonstrate the real application for bed-side detection of the causative pathogen in the early-stage patient populations.
The article is well written but requires additional experiments to support the claims and improve the quality of the work. The paper also has typos that require additional attention before it could be published.
Major comments:
- Since the power of the study was to implement the technology to bed-side and in a hospital environment, a clear comparison of a few reference samples are necessary to demonstrate the implication of testing conditions. A clear comparison of the measured values obtained from a fresh sample vs the stored samples is necessary. Although the authors have indicated the possible issues in the discussion section, it is necessary to identify the discrepancies in the values that may arise due to the different measurement conditions and the sample status.
- How many times the sensors could be used? Whether the purging could be an issue and cause deviation in the analysis over time. Can you store the sensor for several days for multiple uses?
- It is highly recommended to show the readings with a few standard samples and as a function of different concentrations to identify the sensitivity of the measurements in the same settings.
The document has several typos and it is advised that the authors edit the entire article. I am indicating a couple of them below:
- Change the title to “Smell- Adding a New Dimension to Urinalysis”.
- Add the postal code for all the affiliations.
- Abstract: Add a sentence explaining the importance of the study and the key difference from all the earlier studies.
- On page 1 Line 28: Change the sentence to “growth versus with UTI”.
- On page 2 Line 50: Change the sentence to “days to identify the type of bacteria”.
- On page 4 Line 148: Expand the table title.
- Arrange the figure panels (Figure 1a and 1b) into a single file and in one place. Write a figure title and description explaining each panel.
- The text on lines 163-165 (Page 6) is identical to lines 169-171 (Page 7).
- Follow a similar decimal notation for the numbers - 0.003 instead of 0,003.
Reviewer 3 Report
The manuscript entitled: Smell - adding a New Dimension to Urianalysis by Visser et al describe a potential use of the eNose technology as an complementary tool in the diagnostic work-up of urinary tract infections in children.
Taking into account the current state of art related to this interesting topic I would suggest some questions/explanations to analyzed:
Introduction part
- There is a complete research work (Reference 15) about this topic and there will be important to explain the main difference between the aim of the authors and this work from 2014. They clear discrimated the bacaterial composition using an extense number of samples and control samples.
Materials and methods:
- It would be important to use control samples to asses the results that the authors obtained in this study.
Results and discussion:
- It is necessary to explain in detail the scatterplot (Figures 1a and 1b) and improve the image quality of these graphs. It is not clear the importance of this analysis during the discussion.
- Moreover, the results were shown using points and commas indistinctly, please check them (VOC analysis first part).
Conclusion
- After all these points I suggest to improve the conclusion part with a more accurate comparative analysis between the state of art and the authors development.
Reviewer 4 Report
This manuscript showed the potential of eNose as a complementary tool for the urinalysis. The study was well designed and the paper was well structured and written. It is particularly appreciated that the information on patient characteristics and comorbidity were given and associated to the study.
Here are some minor modifications to made before publication:
- page 1, Line 28 and 29, correct these numbers by replacing commas by dots "AUC 0,80; p = 0,009, sensitivity 0,79, specificity 0,75". Same thing for page 5, Line 154-156 "Urine without bacterial growth could be discriminated from urine with bacterial growth (AUC 0,71; p = 0,033, sensitivity 0,64, specificity 0,79). The diagnostic accuracy of VOCs improved when comparing UTI versus urine without bacterial growth (AUC 0,80; p = 0,009, sensitivity 0,79, specificity 0,75).".
- page 3, Line 115, 116, and 117, mistakes in "bacteria species with 103 - 104 CFU/ml", please change to "bacteria species with 103 - 104 CFU/ml".
- page 3, Line 132, "Performance of the eNose was assessed in terms of sensitivity and specificity and by calculating the area under the receiver-operating characteristic (ROC) curve." Please give more details/information how sensitivity and specificity was determined based on experimental data, since they are the important characteristics for eNose.
- The quality of Figure 1a and 1b must be improved: 1) image is blurred; 2) Axis X and its legend are missing; 3) check the number used for Y axis: commas or dot?4) Curve information are cut and not complete.
- A short discussion based on Figure 1 should be added in the discussion part of the manuscript.
Round 2
Reviewer 1 Report
the improvements are significant, ans I recommend publication, however, still I would be happy, if you could use stronger different line style of the presentation of the figure1 : not just blue and green, but may be different symbols and line style for the two different populations.
Reviewer 2 Report
The authors have made necessary changes but was unable to support with additional experiments with the current pandemic situation. I would recommend the authors to add them in the form of additional notes or in their follow up publications.
Reviewer 3 Report
The authors make all the corrections, they improved and explained the figures. Moreover, the authors add one paragraph in order to emphasize the importance of this research (conclusion).
I recommend to accept the manuscript in the present form.
